# Unilateral EMG-Guided Botulinum Toxin for Retrograde Cricopharyngeus Dysfunction: A Prospective Clinical and Neurophysiological Study

**DOI:** 10.3390/toxins17090458

**Published:** 2025-09-12

**Authors:** Giuseppe Cosentino, Chiara Zaffina, Clara Zoccola, Mauro Fresia, Sara Merli, Simone Mauramati, Giulia Bertino, Massimiliano Todisco, Shayan Dodge, Sami Barmada, Enrico Alfonsi, Cristina Tassorelli

**Affiliations:** 1Department of Brain and Behavioral Sciences, University of Pavia, Campus della Salute, presso Policlinico San Matteo, Viale Golgi 19, 27100 Pavia, Italy; cristina.tassorelli@unipv.it; 2Translational Neurophysiology Research Section, IRCCS Mondino Foundation, 27100 Pavia, Italy; chiara.zaffina01@universitadipavia.it (C.Z.); clara.zoccola01@universitadipavia.it (C.Z.); mauro.fresia@mondino.it (M.F.); sara.merli02@universitadipavia.it (S.M.); massimiliano.todisco@mondino.it (M.T.); enrico.alfonsi1@mondino.it (E.A.); 3Department of Otolaryngology Head Neck Surgery, IRCCS Policlinico San Matteo Foundation, 27100 Pavia, Italy; simone.mauramati@gmail.com (S.M.); g.bertino@smatteo.pv.it (G.B.); 4Department of Energy, Systems, Territory and Construction Engineering (DESTEC), University of Pisa, 56126 Pisa, Italy; shayan.dodge@ing.unipi.it (S.D.); sami.barmada@unipi.it (S.B.); 5Headache Science and Neurorehabilitation Center, IRCCS Mondino Foundation, 27100 Pavia, Italy

**Keywords:** cricopharyngeal muscle, retrograde cricopharyngeus dysfunction, botulinum toxin, electromyography, swallowing, upper esophageal sphincter, treatment outcome, dysphagia, burping

## Abstract

Retrograde cricopharyngeus dysfunction (R-CPD) is a recently recognized condition characterized by the inability to burp, typically accompanied by gurgling noises, bloating, and flatulence. Percutaneous botulinum neurotoxin (BoNT) injection into the cricopharyngeus muscle is a minimally invasive treatment with promising effects, although current evidence remains limited. In this prospective, open-label study, we evaluated the clinical effects of increasing doses (10 to 30 U) of EMG-guided unilateral BoNT injection in 67 patients with R-CPD. Symptom severity and quality of life were assessed at baseline and at 1 and 4 months post-treatment. The electromyographic (EMG) parameters of the cricopharyngeus were recorded to explore their association with symptom burden and treatment response. At a 1-month follow-up, 55.2% of patients were classified as responders (satisfaction score ≥ 6/10), with a higher rate (64.4%) observed at higher doses, particularly in female patients. Both symptom severity and quality of life improved significantly at 1 month and were sustained at 4 months. Higher cricopharyngeus EMG activity was associated with more severe symptoms and lesser treatment responses. Unilateral EMG-guided BoNT injection is a safe and effective treatment for R-CPD. Further studies should explore the potential role of electromyography in clarifying the pathophysiological aspects of R-CPD and guiding treatment.

## 1. Introduction

Retrograde cricopharyngeus dysfunction (R-CPD) is a recently described condition characterized by an inability to belch due to inadequate relaxation of the cricopharyngeus muscle during retrograde air flow [1,2,3]. This dysfunction results in symptoms such as gurgling noises in the throat, chest pain or discomfort, abdominal bloating, and flatulence, which can significantly impair patients’ quality of life [4,5,6,7]. First systematically described by Bastian et al. in 2019 [1], who also proposed botulinum neurotoxin (BoNT) injection as a targeted treatment, R-CPD has often been misdiagnosed as gastroesophageal reflux disease (GERD), irritable bowel syndrome (IBS), functional dyspepsia, or aerophagia, highlighting the need for improved diagnostic criteria and targeted therapeutic strategies [1,7,8]. Indeed, a distinctive feature of R-CPD is the high rate of self-diagnosis through social media platforms such as Reddit (r/noburp), YouTube, and TikTok, where many individuals first recognize their symptoms and seek confirmation [4,9]. This phenomenon, while pointing to the growing influence of online communities in shaping disease awareness and clinical pathways, underscores that R-CPD remains poorly defined and underdiagnosed, with no standardized clinical scales available to assess symptom severity or guide longitudinal follow-up.

The current standard of care consists of BoNT injections into the cricopharyngeus muscle, which have traditionally been delivered endoscopically under general anesthesia in doses ranging from 50 to 100 U [1,10,11,12,13,14,15]. More recently, electromyography (EMG)-guided percutaneous injection has emerged as a less invasive alternative [14,16,17,18,19,20]. While case series have reported promising outcomes, systematic studies comparing different injection techniques or testing varying doses to assess potential dose-related effects on treatment response are still lacking. Doruk and Pitman [19] directly compared unilateral percutaneous injection (30 U) to bilateral endoscopic administration (80 U), reporting favorable outcomes with the percutaneous approach, though the endoscopic method remained more effective overall. Notably, the side-effect profile was similar, with dysphagia being more frequent in the higher-dose endoscopic group and hoarseness occurring only in the percutaneous group.

Electrophysiological recordings of the cricopharyngeus muscle during swallowing are routinely used to confirm needle placement during EMG-guided injections, both for the treatment of neurogenic dysphagia and R-CPD [16,18,19,21,22,23,24]. Although these recordings reflect anterograde (i.e., swallowing-related) function rather than retrograde activities like belching, they may still offer indirect insights into upper esophageal sphincter control. Typical swallowing-related EMG patterns include a brief pause in tonic activity, occurring during bolus passage, often preceded by a foreburst (i.e., a phasic increase superimposed on the baseline tonic activity) and followed by a post-pause rebound or squeezing pattern [21,22,24].

This open-label prospective study was designed to enhance the clinical and neurophysiological understanding of R-CPD and to evaluate the role of minimally invasive percutaneous treatment. Specifically, it aims to quantitatively assess changes in the severity of eight core R-CPD symptoms, as well as treatment safety and tolerability, at 1 and 4 months following EMG-guided unilateral percutaneous BoNT injection. Additionally, this study examines changes in quality-of-life parameters and compares outcomes between two BoNT dose groups (lower dose: 10–20 U; higher dose: 25–30 U). It also explores sex-based differences in treatment response and investigates whether baseline cricopharyngeus EMG patterns during saliva swallowing are associated with symptom profiles or clinical outcomes.

By integrating structured clinical assessment with neurophysiological data, this study contributes to a better understanding of R-CPD and informs the clinical use of minimally invasive percutaneous treatment approaches.

## 2. Results

### 2.1. Patient Characteristics

A total of 67 patients with R-CPD were enrolled between March 2023 and May 2025 (mean age: 29.5 ± 7.7 years; range: 15–59; 40 F/27 M). The majority were Italian, with eight individuals from other countries (Bulgaria, Croatia, Greece, Spain, Hungary, Pakistan, Peru, and China).

As assessed at baseline using a structured questionnaire (Table 1, Appendix A), the majority of patients (84%) reported a complete inability to burp, while the remainder experienced only incomplete and effortful burping. Symptoms such as flatulence, gurgling noises, abdominal bloating, and postprandial chest discomfort were highly prevalent, affecting over 90% of participants. Other frequent features included painful hiccups (88%) and increased hiccup frequency (66%). Quality-of-life (QoL)-related impairments were also common: 85% of patients reported some degree of food or beverage avoidance (particularly carbonated drinks, foods that may trigger or worsen symptoms, or meal skipping), and 76% avoided social situations to varying extents, either to prevent symptom exacerbations or due to ongoing symptoms. The perceived impact on QoL was relevant, with 79% of patients rating it as ≥ 8 on a 0–10 scale. Notably, the vast majority of patients presented to our clinic after self-identifying the condition through online resources. In most cases, they recognized their own symptoms through testimonies shared by other individuals on social media platforms, particularly TikTok, rather than being referred by healthcare professionals. Only a small subset of patients had been referred to us by specialists. Detailed clinical and anamnestic characteristics of the 67 enrolled patients are summarized in Table 2, while Table 3 provides an overview of prior medical evaluations, diagnostic investigations, and treatments received.

All participants completed the baseline assessment and 1-month follow-up. Twenty-two patients received a 10–20 U dose of BoNT (lower-dose group; mean age: 29.9 ± 8.3 years; 11 F/11 M), and 45 patients received a 25–30 U dose (higher-dose group; mean age: 29.2 ± 7.6 years; 29 F/16 M). There were no significant differences in age or sex distribution between these two groups.

At the time of statistical analysis, 40 patients had completed the 4-month follow-up (mean age: 29.6 ± 8.0 years; 21 F/19 M). Of these, 20 patients had received the 10–20 U dose (mean age: 29.5 ± 8.6 years; 9 F/11 M), and 20 had received the 25–30 U dose (mean age: 29.6 ± 7.5 years; 12 F/8 M). No significant differences were observed between the two dose groups in terms of age or sex distribution.

### 2.2. Responder Rate at 1 Month and Stratified Analysis by Dose and Sex

Restoration or facilitation of the ability to burp was achieved in 62 out of 67 patients (92.5%) within the first week following treatment, indicating a high immediate success rate. At the 1-month follow-up, 37 out of 67 patients (55.2%) reported satisfaction scores of ≥6/10 and were classified as treatment responders, while the remaining 30 patients (44.8%) were considered non-responders. Stratification by treatment dose revealed a significantly higher proportion of responders in the 25–30 U group, with 29 out of 45 patients (64.4%) classified as responders, compared with only 8 out of 22 patients (36.4%) in the 10–20 U group. This difference was statistically significant (χ^2^ = 4.71, *p* = 0.03), suggesting a dose-dependent effect on subjective treatment satisfaction.

When further stratified by sex, the dose–response relationship was detected only in the female group, where 20 out of 29 treated with the higher 25–30 U dose (69%) were classified as responders vs. only 3 out of 11 (27%) in the lower 10–20 U group. This difference was statistically significant (χ^2^ = 5.67, *p* = 0.02). Among male patients, the responder rate was 56.2% (9/16) in the 25–30 U group and 45.5% (5/11) in the 10–20 U group, with no significant difference observed between groups (χ^2^ = 0.30, *p* = 0.58).

### 2.3. Symptom Severity and Quality of Life: Longitudinal Changes at 1 and 4 Months

Among the 40 patients who completed all three assessments (baseline, 1 month, and 4 months), including individuals from both the 10–20 U and 25–30 U dose groups, a significant and sustained reduction in overall symptom severity was observed over time (Table 4). The Friedman test revealed a highly significant effect across timepoints for the total eight-item symptom score (χ^2^ = 44.8, *p* = 0.00001). Post hoc Wilcoxon signed-rank tests, corrected for multiple comparisons, confirmed significant improvement from baseline to both the 1-month (*p* = 0.00001) and 4-month follow-up (*p* = 0.00001), while no significant difference was found between the two follow-ups, indicating that the therapeutic benefit achieved at 1 month was maintained at 4 months. Each individual symptom showed a similar pattern of improvement.

Quality of life (QoL) domains, including global impact, food/beverage avoidance, and social avoidance, also improved significantly over time (Table 4). Each individual domain followed the same pattern: significant reductions between baseline and both follow-up timepoints (*p* = 0.00001), with no significant differences between 1 and 4 months.

### 2.4. Dose-Related Differences in Symptom Change at 4 Months: Item-Level Analysis

To assess the sustained clinical impact of BoNT and better explore dose-related effects, we conducted a non-parametric item-level analysis of symptom score changes (delta: baseline to 4-month follow-up) between the lower-dose (10–20 U) and higher-dose (25–30 U) groups (Table 5). This analysis included the 40 patients with complete 4-month data and, despite the smaller sample compared with the 1-month follow-up, was selected to emphasize long-term outcomes and treatment duration.

In the overall cohort, the patients in the higher-dose group tended to show greater reductions in symptom severity across most of the eight core R-CPD symptoms and QoL items. However, these differences did not reach statistical significance for any individual item (all *p* > 0.05), suggesting a consistent but variable clinical trend favoring the higher dose.

When the analysis was stratified by sex, more distinct patterns emerged. Among female patients, the higher-dose group showed significantly greater improvement in bloating, the total 8-item symptom score, and treatment satisfaction (*p* < 0.05 for each). These findings align with the previously observed higher responder rate in women treated with 25–30 U and suggest a sex-specific dose–response effect that remains evident over time. Conversely, no significant differences were observed between dose groups in male patients for any symptom item, indicating a more homogeneous response profile in men regardless of dose.

### 2.5. Adverse Effect Severity and Duration

At least one adverse effect was reported by each of 59 out of 67 patients (88%), the most common being transient dysphagia to solid foods (Table 6). Importantly, no cases of severe adverse reactions were reported, and no patients required hospitalization. The median severity score across the cohort was 1 (i.e., minimal; IQR: 1–2), and the median duration of adverse effects was 30 days (IQR: 14–30). In most cases, symptoms subsided within the first two weeks, although minimal residual effects could persist for a few additional weeks.

A Mann–Whitney U test was conducted to compare adverse effect severity and duration between the two dose groups. In the 10–20 U group, the median severity was 1 (IQR: 1–2) and the median duration was 30 days (IQR: 14–30). In the 25–30 U group, the median severity was 2 (i.e., mild; IQR: 1–3) and the median duration was 30 days (IQR: 21–30). However, these differences were not statistically significant for either severity (*p* = 0.13) or duration (*p* = 0.38).

When stratified by sex, female patients treated with the higher dose reported significantly greater severity of adverse effects (median: 2; IQR: 2–3) compared with those who received the lower dose (median: 1; IQR: 1–2) (*p* = 0.03). In contrast, no significant differences were observed in the duration of adverse effects between males and females.

### 2.6. Electrophysiological Correlations and Predictors of Treatment Response

Correlation analyses were conducted to examine associations between the baseline electrophysiological parameters of the cricopharyngeus muscle and clinical symptomatology. A summary of these electrophysiological parameters for the full cohort is provided in Table 7. While no significant relationships were found with the total eight-item symptom score, several item-level correlations emerged. Greater difficulty in burping was significantly associated with higher tonic basal activity of the cricopharyngeus muscle in terms of both peak (r = 0.28, *p* = 0.02) and mean amplitude (r = 0.29, *p* = 0.02), as well as with increased mean amplitude (r = 0.26, *p* = 0.03) and area under the curve (r = 0.25, *p* = 0.04) during the EMG pause related to swallowing. Additionally, reduced burp frequency, reflected by higher symptom scores, was significantly associated with increased peak tonic basal activity (r = 0.25, *p* = 0.04). The EMG area during basal activity also correlated positively with the severity of gurgling noises (r = 0.26, *p* = 0.04). Furthermore, the duration of the squeezing phase was significantly associated with flatulence frequency (r = 0.28, *p* = 0.02).

To determine whether electrophysiological measures could predict clinical responses, we analyzed correlations between baseline EMG values and changes in symptom severity from baseline to the 1-month follow-up (i.e., delta scores). We found that higher maximum squeezing amplitudes were negatively correlated with improvements in chest pain/discomfort (r = −0.26, *p* = 0.03), hiccup frequency (r = −0.32, *p* = 0.01), and painful hiccups (r = −0.25, *p* = 0.04). Similarly, negative correlations with hiccup frequency were observed for the mean squeezing amplitude (r = −0.32, *p* = 0.01) and squeezing area (r = −0.47, *p* = 0.001). These findings suggest that increased cricopharyngeus muscle activation during the squeezing phase may be associated with a less-favorable reduction in certain symptoms. We also compared the baseline EMG values between responders and non-responders, defined as those reporting satisfaction scores of ≥ 6 at the 1-month follow-up. Non-responders exhibited significantly higher peak amplitudes (*p* = 0.02), mean amplitudes (*p* = 0.04), and areas under the curve (*p* = 0.02) during the EMG pause. This pattern may reflect reduced cricopharyngeus relaxation during saliva swallowing, potentially indicating baseline hyperactivity of the muscle that could hamper the efficacy of BoNT treatment.

To further explore whether demographic and treatment-related variables, together with electrophysiological parameters, could independently predict clinical responses, we conducted logistic regression analyses. In the univariate models, sex (OR 1.25; *p* = 0.65) and age (OR 1.02; *p* = 0.40) were not associated with treatment outcome, while dose showed only a non-significant trend (OR 1.07; *p* = 0.07). In multivariate models adjusted for sex, age, and dose, several electrophysiological parameters emerged as significant independent predictors of treatment response. Specifically, lower cricopharyngeus EMG pause peak amplitudes (OR 0.99; *p* = 0.04), pause mean amplitudes (OR 0.99; *p* = 0.02), pause areas under the curve (OR 0.99; *p* = 0.03), and squeezing mean amplitudes (OR 0.99; *p* = 0.04) were associated with a higher likelihood of response. Noteworthily, dose also reached statistical significance in models that included either the cricopharyngeus EMG pause peak amplitude (OR 1.08; *p* = 0.04) or squeezing mean amplitude (OR 1.11; *p* = 0.02) while remaining non-significant in univariate analysis.

No significant differences in baseline EMG parameters were observed between male and female patients, nor were any differences found between patients in whom the muscle was treated on the left versus the right side.

### 2.7. Additional Correlation Analyses

Additional correlation analyses were conducted to evaluate whether baseline symptom severity (assessed through the 8 core items and the 3 QoL-related items) was associated with patient age at the time of evaluation or with symptom changes over time. No significant correlations were found between age and baseline scores or delta scores (baseline to 1 and 4 months), with the exception of gurgling noises, which showed a weak but statistically significant negative correlation with age (r = −0.26, *p* = 0.04), indicating greater symptom severity in younger patients.

## 3. Discussion

To our knowledge, this is the first study to integrate structured clinical outcome measures with electrophysiological data to explore both the therapeutic effects and underlying physiological mechanisms in R-CPD. Our findings confirm and expand upon the previous literature supporting the safety and efficacy of EMG-guided BoNT injection into the cricopharyngeus muscle [14,16,17,18,19,20] while also examining the influence of dosage and sex on clinical response.

### 3.1. Clinical Efficacy and Dose Response

Our findings demonstrate a significant and sustained reduction in symptom severity, with marked improvements evident at 1 month and maintained at the 4-month follow-up. Benefits were consistent across the eight core R-CPD symptoms and key QoL measures. A clinically meaningful dose–response effect also emerged: while both dose ranges (10–20 U and 25–30 U) were effective, the higher dose was associated with a higher response rate and greater symptom reduction, particularly in female patients, who showed significantly greater improvements in abdominal bloating, total symptom burden, and treatment satisfaction. The absence of a clear dose–response effect in male patients may indicate the need for a higher dose threshold to achieve similar benefits. This aligns with previous evidence from cervical dystonia and aesthetic treatments, where men typically require higher botulinum toxin doses to reach comparable outcomes [25,26]. Alternatively, bilateral percutaneous injection, either first-line or second-step, may prove more effective in male patients.

Noteworthily, regression analyses further confirmed the independent role of the dose: although dose alone showed only a trend in univariate analysis, it reached statistical significance in multivariate models when considered together with specific electrophysiological predictors, namely cricopharyngeus EMG pause peak amplitude and squeezing mean amplitude. This finding suggests that the injected BoNT dose contributes to clinical outcome beyond the effect of baseline neurophysiological characteristics.

Comparisons across studies remain challenging due to variations in follow-up duration, outcome measures, injection techniques, and toxin formulations. In our cohort, 92.5% of patients regained the ability to burp within the first week after treatment, in line with the findings of Lichien et al. [7]. This parameter mainly reflected the early pharmacological effect of BoNT on cricopharyngeus relaxation and was often incomplete or not yet perceived as satisfactory. Nevertheless, this early improvement is clinically relevant, as it supports the diagnosis of R-CPD. However, it did not always translate into a broader and sustained benefit across the full symptom spectrum, which explains the lower responder rate observed at the 1-month evaluation when assessed through the structured questionnaire. The sustained clinical responses observed at 1 and 4 months appear lower than those reported in most studies using endoscopic injection, which typically involves bilateral delivery and higher toxin doses. Regarding EMG-guided percutaneous approaches, our 1-month responder rate of 64.4% in patients receiving 25–30 U closely mirrors the 64.9% reported by Doruk and Pitman [19] with a unilateral injection of 30 U. Wajsberg et al. [16] used a higher-dose bilateral percutaneous approach, reporting greater efficacy but also breathing-related side effects. Similarly, Mailly et al. [14] reported a higher (84%) response rate following a single unilateral injection of 75 U. These findings might suggest that higher doses could enhance efficacy, although the impact on tolerability remains uncertain. Therefore, further targeted studies are needed to optimize both dosing and injection strategies, also in relation to clinical and demographic parameters that may influence treatment response and tolerability.

### 3.2. Safety Profile and Practical Implications

Side effects occurred frequently in our population, but their intensity was generally minimal or mild and self-limiting. Most symptoms resolved within the first two weeks, although minimal residual complaints could persist slightly longer. The most commonly reported adverse effect was transient dysphagia to solid foods. In all cases, patients were able to manage this side effect successfully by adopting simple compensatory strategies, such as avoiding hard or dry foods, chewing thoroughly, swallowing in small bites, tilting the head slightly forward while swallowing, and following each bite with a small sip of water.

While patients receiving higher doses (25–30 U) reported greater side-effect severity, consistent with the suggestion that higher doses may be associated with reduced tolerability [19], this difference was not statistically significant overall. However, stratified analysis revealed that female patients treated with higher doses experienced a significantly higher prevalence of moderate, as opposed to mild or minimal, side effects, suggesting a possible sex-related sensitivity to the BoNT. Nonetheless, the use of the higher dose in female patients appears justified, given the significantly greater clinical benefit and only a modest, clinically acceptable increase in side-effect severity. Importantly, future studies with larger cohorts are warranted to more systematically explore whether specific subgroups, potentially defined by sex, symptom burden, or body mass, may exhibit different benefit-to-risk profiles at varying doses.

### 3.3. Cricopharyngeus EMG Parameters and Clinical Correlations

This study provides the first integrated analysis of symptom profiles and quantitative cricopharyngeus EMG data in R-CPD.

Notably, higher tonic basal activity of the cricopharyngeus muscle was significantly associated with greater difficulty in burping, suggesting that increased resting muscle tone may contribute to impaired retrograde airflow, one of the hallmark features of R-CPD. In addition, greater EMG activity during the swallowing-related pause was also correlated with burping difficulty. This pattern may reflect a reduced capacity of the cricopharyngeus muscle to undergo appropriate reflex relaxation, potentially indicating tonic hyperactivity or impaired inhibitory control, also in line with findings from high-resolution esophageal manometry studies [8,13,27]. While our data primarily reflect swallowing-related function, it is plausible that similar deficits may also affect retrograde relaxation mechanisms involved in belching, though this remains a hypothesis requiring targeted investigation.

Interestingly, reduced burp frequency was significantly associated with greater peak tonic activity, supporting the notion that tonic overactivity of the cricopharyngeus muscle may contribute to impaired belching. Basal EMG activity was also positively correlated with the severity of gurgling noises, suggesting that sustained resting tone may promote upper esophageal air retention and the generation of audible vibratory phenomena. In addition, we observed a significant association between the duration of the squeezing phase and the frequency of flatulence. This might suggest that prolonged cricopharyngeus contraction could play a role in facilitating antegrade air transit through the esophagus, possibly contributing to more effective esophageal clearance. However, this remains speculative, and studies combining EMG with high-resolution esophageal manometry would be needed to investigate whether increased squeezing activity is meaningfully associated with esophageal peristalsis and air elimination dynamics.

When examining predictive markers of therapeutic response, we found that responders, compared to non-responders, had significantly higher peak and mean amplitudes, as well as greater areas under the curve during the EMG pause phase. These results point to a reduced degree of cricopharyngeus relaxation during swallowing in non-responders, consistent with a persistent hypertonic state that could hamper the efficacy of BoNT. In line with this, we also observed that higher baseline squeezing activity was negatively associated with clinical improvement at the 1-month follow-up, with specific associations observed for chest pain/discomfort, increased hiccup frequency, and painful hiccups. Moreover, logistic regression analyses confirmed that specific electrophysiological markers, namely lower cricopharyngeus EMG pause peak amplitude, pause mean amplitude, pause area under the curve, and squeezing mean amplitude, were independent predictors of clinical response. Importantly, these associations remained significant even after adjustment for demographic variables and dose, underscoring the role of baseline EMG features as potential objective biomarkers of treatment efficacy. Taken together, these findings suggest that increased cricopharyngeus muscle activation may represent a less responsive muscle profile, possibly requiring more intensive or tailored therapeutic strategies.

Based on these preliminary results, future studies are needed to assess whether specific EMG features at baseline could serve as reliable markers to guide treatment decisions.

It is important to emphasize that this study specifically focused on EMG assessment of the cricopharyngeus muscle during tonic activity and spontaneous saliva swallowing, primarily to identify correct needle placement. We did not evaluate the muscle’s responsiveness to retrograde stimuli, such as esophageal distension, using challenge paradigms like those recently applied in high-resolution manometry with carbonated water [28]. While such tests may have provided valuable and more direct insights into retrograde reflexes, incorporating these challenge paradigms would have required protocol modifications and extended recording sessions, which were beyond the intended scope of this study.

### 3.4. Strengths and Limitations

This study represents an early attempt to systematically characterize the clinical and electrophysiological features of R-CPD in a prospective framework. Strengths include the use of a disease-specific structured questionnaire incorporating both symptom severity and quality-of-life metrics in alignment with recently proposed outcome reporting standards [6]. Electrophysiological data were quantitatively analyzed to support an objective assessment of cricopharyngeus muscle activity, and analyses stratified by dose and sex allowed for the identification of potentially meaningful clinical subgroups.

Nonetheless, several limitations must be acknowledged. This study employed a non-randomized design with sequential dose escalation, which could introduce temporal or selection-related biases. Post-treatment EMG recordings were not performed, limiting our ability to directly assess the physiological impact of BoNT injection on muscle function. While the 1-month outcome data were complete for all patients, a portion of the cohort had not yet reached the 4-month follow-up at the time of analysis, which may have reduced the statistical power of long-term comparisons. Additionally, the absence of a placebo control group limits causal inference, although implementing placebo injections in R-CPD would be ethically problematic given the documented high efficacy of BoNT in this population. Finally, normative EMG data from healthy individuals are lacking, which constrains the interpretation of electrophysiological values in absolute terms. This limitation is also difficult to overcome, as performing needle EMG of the cricopharyngeus in healthy volunteers would raise ethical concerns.

## 4. Conclusions

In conclusion, we confirm previous observations that EMG-guided unilateral BoNT injection into the cricopharyngeus muscle is a safe, effective, and well-tolerated intervention for patients with R-CPD [14,16,17,18,19,20]. The treatment leads to important and sustained reductions in symptom severity, with higher doses showing enhanced benefits in female patients. Adverse effects were generally mild and transient.

The electrophysiological findings from this study may provide indirect insights into the pathophysiology of R-CPD, suggesting that distinct EMG parameters could be associated with specific symptom domains and differential treatment outcomes. Higher tonic basal activity and reduced relaxation during the swallowing-related EMG pause were associated with more severe impairment in belching and lower treatment responses, pointing to a hypertonic or less modulable muscle profile. Conversely, increased squeezing activity was related to symptoms such as flatulence and was also linked to poorer improvement in certain upper esophageal symptoms. Noteworthily, logistic regression analyses confirmed that both dose and specific EMG parameters independently predicted treatment response, highlighting their potential as objective biomarkers to guide therapeutic decision-making. These findings lay the groundwork for future studies integrating EMG with high-resolution manometry to better elucidate underlying mechanisms and support the development of individualized treatment strategies. Further exploration of sex-related differences, dose optimization, and long-term clinical outcomes will also be essential. In addition, comparative studies are needed to evaluate different procedural techniques, such as percutaneous EMG-guided versus endoscopic injection, unilateral versus bilateral approaches, and variations in total doses or numbers of injection sites, to determine which strategies yield the best risk–benefit profile for individual patients.

A key area for future research involves understanding the duration of treatment effects over time. It remains unclear how many patients will require retreatment and at what intervals, or how to best manage those who show suboptimal responses. In our clinical experience, we have recently begun offering contralateral reinjection with 30 U of BoNT to patients who either failed to respond or experienced partial loss of effect within a few weeks or months after first treatment. In our experience, the preliminary results from this staged bilateral approach appear promising and warrant systematic investigation in future studies.

Ultimately, a more nuanced understanding of upper esophageal sphincter neuromotor control and its interplay with esophageal motility will be essential for improving diagnostic accuracy and optimizing therapeutic strategies in R-CPD.

## 5. Materials and Methods

### 5.1. Subjects

This prospective study enrolled consecutive patients diagnosed with R-CPD at our center between March 2023 and April 2025. Due to the limited epidemiological data on R-CPD and the exploratory nature of this non-sponsored study, a formal sample size calculation was not feasible. At study initiation, the number of potentially eligible patients was unknown, and the final sample size was determined based on actual patient availability.

Patients were eligible for inclusion if they presented with complete or near-complete inability to burp, along with at least two of the following: (1) abdominal bloating or chest pain, especially after eating; (2) gurgling noises from the chest or lower neck; and (3) excessive flatulence [1,3,5]. Patients who had not yet undergone gastroenterology or otolaryngology evaluation were referred to specialists before potential inclusion in this study.

Patients were excluded from this study if they had neurological, neuromuscular, or structural conditions affecting the upper esophageal sphincter, including those with oropharyngeal dysphagia and known esophageal motility disorders, such as hypercontractile esophagus or achalasia. Although patients with overt dysphagia were excluded, we included those reporting occasional mild dysphagic symptoms (e.g., pharyngeal lump sensation or slight food passage difficulty). These symptoms were generally not spontaneously mentioned by patients whose overall clinical presentations were otherwise fully consistent with R-CPD, but they were usually reported when specifically queried about. In such cases, inclusion was permitted only after comprehensive gastroenterological or ENT evaluation (supported by fiber-optic endoscopic evaluation of swallowing, FEES; videofluoroscopic swallowing study, VFSS; or high-resolution manometry, HRM) had ruled out other identifiable causes of dysphagia.

Individuals with bleeding disorders or those with contraindications to botulinum toxin use (e.g., myasthenia gravis) were also excluded. Additionally, patients with severe psychiatric conditions that could have interfered with the interpretation of symptoms were not considered eligible. Patients with previous diagnoses of ineffective esophageal motility (IEM) confirmed by esophageal manometry were not excluded from this study, as this condition appears to be frequently associated with R-CPD and does not contraindicate treatment with botulinum toxin [8,13]. The study flow, including patient enrollment, group assignment, and follow-up, is summarized in Figure 1.

Given the exploratory nature of this study and the inclusion of patients undergoing what is currently considered the treatment of choice for R-CPD, the intervention (EMG-guided botulinum toxin injection) was not administered as part of a predefined research protocol but rather as part of standard clinical practice. Although data collection followed a prospective design with structured follow-up, treatment allocation was not randomized, and no experimental procedures were introduced. The decision to analyze the collected data for publication purposes was made retrospectively. Accordingly, this study does not meet the ICMJE definition of a clinical trial requiring mandatory pre-registration.

### 5.2. Clinical Assessment and Questionnaire Administration

All patients completed a structured, self-administered questionnaire at baseline, developed through a comprehensive literature review and refined with input from two healthcare professionals with personal experience of R-CPD (Appendix A). Their contributions ensured that the content was both clinically relevant and reflective of patient experiences. The questionnaire was designed to comprehensively assess symptom severity and its impact on daily life. It consisted of an initial section with eight core symptom items, followed by three additional items evaluating the impact on behavioral and social functioning, as well as overall quality of life. The item assessing perceived impact on quality of life was developed specifically for this study, drawing inspiration from global assessment tools commonly used in clinical research, with the aim of assessing each patient’s subjective experience in a simple and intuitive format. A segmented horizontal line numbered from 0 (no impact) to 10 (extremely severe impact), as shown in Table 1, was used for scoring, in accordance with approaches that link numeric rating scales to patient-perceived meaningful change and the use of global scales in clinical practice [29,30].

Each item was scored for severity, with the first eight items contributing to a total score ranging from 0 to 28 and the additional three items contributing up to 18 points for a total maximum score of 46. Higher scores indicated greater symptom burden and impact.

The questionnaire was preliminarily tested for usability and comprehension in a small group of individuals with suspected R-CPD. Informal test–retest procedures conducted in a subset of participants demonstrated stable and consistent responses over time. Although formal psychometric validation was not performed, the tool appeared well-aligned with the clinical presentation and subjective experience of the patients.

In addition, detailed clinical history and qualitative information were collected at baseline, including symptoms potentially associated with R-CPD (see Table 2), such as emetophobia and dysphagia, as well as comorbid conditions, including GERD and anxiety disorders. Information was also gathered regarding previous specialist evaluations, diagnostic testing, prior failed treatments, and any pre-existing diagnoses (see Table 3).

To monitor symptom evolution and treatment response, the same questionnaire used at baseline (comprising the 8 core items and 3 impact items) was administered again at the 1-month and 4-month follow-up timepoints. The follow-up version additionally included items assessing the presence and intensity of other potentially associated symptoms (as listed in Table 2), as well as patient satisfaction with treatment and the occurrence of any adverse effects.

Given the inherently subjective nature of R-CPD symptoms and the lack of objective follow-up measures, a simple and intuitive tool such as the NRS was considered appropriate for assessing patient-perceived treatment success. Satisfaction with treatment was therefore assessed using an NRS, presented as a horizontal line numbered from 0 (no satisfaction) to 10 (maximum satisfaction). The choice of an 11-point NRS for treatment satisfaction was informed by its extensive use and validation in assessing subjective patient outcomes such as pain intensity [31]. Although validation in direct satisfaction measures in adults is limited, a single-item NRS has demonstrated correlation with multi-item satisfaction scales [32]. In the absence of established cut-off values for treatment satisfaction, a threshold of ≥6 was adopted, reflecting the interpretation that scores of 6 or higher represent a satisfactory level of perceived benefit.

Adverse effects were rated using a 5-point scale (0–4), where 0 indicated no side effects; 1, minimal (barely noticeable, no interference with daily activities); 2, mild (noticeable but not limiting); 3, moderate (interfering with some daily activities and possibly requiring minor intervention); and 4, severe (disabling or significantly disruptive, potentially requiring medical attention). This evaluation was based on each patient’s subjective perception.

### 5.3. Treatment Procedure

Patients diagnosed with R-CPD and without contraindications were treated with unilateral EMG-guided injections of onabotulinumtoxinA, diluted at 100 U per 1 mL of normal saline. We used Ambu Neuroline Inoject needle electrodes (0.50 mm, 25 G). These needles have a stainless-steel shaft covered by an acrylic insulating layer and a silicone outer coating so that only the beveled tip is electrically active while the shaft remains fully insulated, thereby minimizing the risk of EMG signal contamination. The procedure was performed as an off-label use, justified by the available literature data, and carried out after obtaining specific informed consent for off-label treatment. Given the limited knowledge available on percutaneous injection for R-CPD at the time treatment began, we adopted a cautious, stepwise dose-escalation strategy, starting at 10 U and increasing by 5–10 U in successive patients, provided no significant adverse effects were observed. This approach was informed by our prior experience with low-dose percutaneous BoNT injection for neurogenic dysphagia [23,33] and aimed to ensure patient safety in the absence of direct visualization of the cricopharyngeus muscle.

Unlike endoscopic techniques, which allow direct targeting under general anesthesia, the percutaneous approach relies solely on EMG feedback, theoretically increasing the risk of mis-targeting, i.e., inadvertent injection into adjacent structures, such as the inferior pharyngeal constrictor or intrinsic laryngeal muscles. To further minimize this risk, all injections were administered unilaterally, rather than bilaterally as is commonly practiced in endoscopic protocols.

Clinical improvement was observed even with the initial 10 U dose in a proportion of patients, encouraging further exploration of low-dose regimens. Notably, some individuals, such as the first case treated at our center, later published by Pavesi et al. [18], achieved long-lasting remission, with complete symptom resolution sustained for more than two years and ongoing to date. However, the overall response rate at lower doses was lower than that reported in the original R-CPD series by Bastian et al. (2019) [1]. In light of these findings, and in the absence of significant adverse events in our cohort, the dose was gradually increased up to a maximum of 30 U, supported by the emerging literature favoring higher percutaneous doses [19]. Dose assignment followed a sequential pattern, uninfluenced by clinical or demographic characteristics, thereby minimizing selection bias.

All injections were performed by the same physician (G.C.), who had greater manual proficiency on the right side, resulting in a consistent preference for targeting the right portion of the cricopharyngeus muscle. In six cases, the left side was injected instead when right-sided access was less favorable.

BoNT was injected in patients placed in a neutral supine position. The needle was inserted at an angle of approximately 45 degrees, about 1 cm lateral to the palpable edge of the cricoid cartilage, serving as an anatomical landmark. Dry (saliva) swallowing was used to identify the correct target within the cricopharyngeus muscle, confirmed by the presence of baseline tonic EMG activity that paused during swallowing, followed by the characteristic post-swallow tonic rebound (squeezing phase). To avoid misplacement of the needle into adjacent muscles such as the posterior cricoarytenoid or neck flexors, deep inspiration and neck flexion maneuvers were tested under EMG monitoring.

After the injection, patients were monitored on site for at least one hour to evaluate for any immediate adverse effects and to ensure post-procedural safety.

### 5.4. Electromyographic Signal Acquisition and Analysis

In each patient, at least one cricopharyngeus electromyographic (EMG) recording of ≥10 s was obtained during dry swallowing (Figure 2). Each trace was required to include the full sequence of swallowing-related phases: baseline tonic activity, pre-activation (foreburst), EMG pause, squeezing phase, and return to post-swallow tonic activity.

When more than one usable recording was available, data from individual traces were averaged to generate a single representative profile per patient. Electromyographic traces were analyzed offline using a custom Python-based processing pipeline developed by SD (version 3.12.3, Python Software Foundation, https://www.python.org). After excluding low-quality traces (e.g., those with excessive drift, motion artifacts, or insufficient duration), five analysis windows (A–E) corresponding to physiologically relevant EMG phases were manually segmented. Each trace was independently reviewed by two raters (G.C., M.T.), with final segmentation defined by consensus. For each window, the following parameters were computed: duration (ms), mean amplitude (µV), peak amplitude (µV), and normalized area under the curve (µV·s). Pre-swallow tonic activity was used as the baseline reference to minimize post-deglutition artifacts.

### 5.5. Statistical Analysis

All statistical analyses were performed using Statsoft software (version 14.2.0). Continuous variables were expressed as means ± standard deviation (SD) or as medians and interquartile ranges (IQRs), depending on the distribution of the data, while categorical variables were reported as absolute frequencies and percentages.

The primary outcome measure was the proportion of treatment responders, defined as patients reporting a satisfaction score of ≥6 out of 10 at the 1-month follow-up. This analysis was conducted in the full cohort of 67 patients who completed the baseline and 1-month evaluations. The responder rate was further analyzed according to treatment dose (10–20 U vs. 25–30 U dose range) and sex (male vs. female) in order to explore potential dose-related and sex-related differences in subjective treatment satisfaction. Additional primary endpoints included the changes in severity of the eight core R-CPD symptoms (both as a total score and as individual item scores) between baseline and the follow-up visits at 1 and 4 months, as well as changes in QoL parameters over the same timepoints. Longitudinal comparisons of symptom severity and QoL scores were conducted in the subgroup of 40 patients, with complete data across all three timepoints. The Friedman test was used to assess changes over time, followed by Wilcoxon signed-rank tests for post hoc pairwise comparisons. To account for multiple testing, the Bonferroni correction was applied, setting the significance threshold for post hoc comparisons at *p* < 0.0167.

Predefined secondary analyses explored the association between treatment responses and injected doses (lower, 10–20 U vs. higher, 25–30 U dose range), both in the entire cohort and stratified by sex. Changes in symptom severity and QoL measures were compared between dose groups using the Mann–Whitney U test. Adverse events were analyzed in terms of subjective severity (rated 0–4) and duration (measured in days), and their distribution was compared across treatment groups and sexes. In addition, we investigated correlations between baseline electrophysiological features and clinical variables, including both baseline symptom profiles and post-treatment outcomes at 1 month, assessed as delta scores. Finally, baseline EMG measures were compared between responders and non-responders to evaluate their potential as predictors of treatment efficacy.

The normality of the continuous variables was assessed using the Shapiro–Wilk test. As most clinical and neurophysiological variables did not follow normal distribution, non-parametric statistical methods were applied throughout. Categorical variables were compared using the chi-square test, while associations between continuous and ordinal variables were assessed with Spearman’s rank correlation. Logistic regression analyses were performed considering clinical response (responder = 1; non-responder = 0) as the primary outcome. In the first step, univariate models were tested, including sex, age, and dose as independent predictors. In the second step, multivariate models were constructed by including sex, age, and dose together with a single electrophysiological parameter at a time, selected on the basis of preliminary correlation analyses. Results were expressed as odds ratios (ORs) with 95% confidence intervals (CIs).

All statistical tests were two-tailed, with significance set at *p* < 0.05 unless otherwise corrected for multiple comparisons.

## Figures and Tables

**Figure 1 toxins-17-00458-f001:**
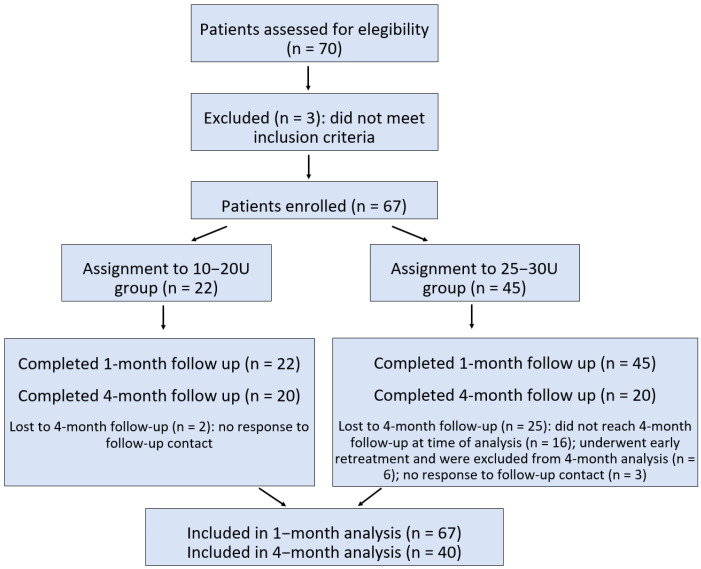
Study flow chart showing enrollment, group allocation, follow-up completion, and the number of patients included in the 1-month and 4-month analyses.

**Figure 2 toxins-17-00458-f002:**
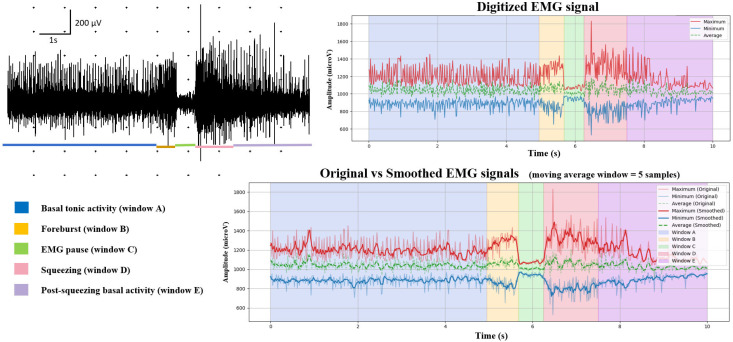
Electromyographic analysis of cricopharyngeus muscle activity during dry swallowing in a representative patient. (**Left**) Raw EMG trace (10 s) with color-coded segmentation of five functional swallowing phases. (**Right, top**) Digitized EMG signal (min, max, average values). (**Right, bottom**) Comparison between original and smoothed signals using a five-sample moving average; analysis windows (A–E) correspond to physiologically defined swallowing phases. EMG data were exported as image files and processed using a custom Python-based pipeline. Following signal boundary selection, axes were calibrated (time: 0–10 s; amplitude: pixel-Y to µV). A smoothing filter was applied to reduce high-frequency noise while preserving the physiological waveform. Signal metrics, including duration, mean and peak amplitude, and area under the curve, were computed within each manually defined window.

**Table 1 toxins-17-00458-t001:** Table 1 summarizes the structured questionnaire used to assess symptom severity and quality of life in patients with retrograde cricopharyngeus dysfunction (R-CPD) at baseline, 1 month, and 4 months post-treatment.

Domain	Items (Summary)	Scoring Range
Core symptoms	Ability to burp Burp frequency Gurgling noises Abdominal bloating Chest discomfort/pain Flatulence Hiccup frequency Painful hiccups	0–3 or 0–4 per item
Quality of life	Avoidance of certain foods/beverages Avoidance of social events Global impact of symptoms on daily life	0–4 per item Numeric rating scale, 0–10
Total score	8 symptom items (0–28) 3 QoL items (0–18) Maximum total: 46	

This tool includes eight core symptom items and three quality-of-life items, with higher scores indicating greater symptom burden and impact. The maximum total score is 46 points (28 from symptom items and 18 from QoL items). A concise summary of domains and scoring ranges is reported here, while the full questionnaire with all questions and response options is provided as Appendix A.

**Table 2 toxins-17-00458-t002:** Baseline clinical and anamnestic characteristics of the 67 patients enrolled in this study.

Clinical and Historical Features	Baseline Findings	Clinical Observations and Outcomes (Referring to % of Those Affected)
Ability to burp	Complete inability to burp: 56/67 patients (83.6%); incomplete burping, achievable only through specific strategies: 11/67 (16.4%)	Onset of inability to burp: 56 patients (84%) do not recall ever being able to burp; 7 (10%) remember being able to burp during childhood, with a gradual onset of difficulty; and 2 (3%) reported a sudden loss of the ability to burp, both associated with the onset of emetophobia (one after an ingestion episode, the other following nausea and vomiting post-anesthesia). Main strategies to facilitate burping: deep breathing, inducing vomiting through manual maneuvers, squatting (Valsalva maneuver), and neck movements or self-massage.
Burp frequency	Never: 53/67 (79%); rarely: 14/67 (21%)	Fifty-two patients (78%) reported increases in burp frequency 1 month after treatment.
Gurgling noises	64/67 (95.5%)	Fifty-three patients (83%) reported reduction in the frequency of gurgling noises 1 month after treatment.
Abdominal bloating	62/67 (92.5%)	Forty-seven patients (76%) reported reduction in abdominal bloating 1 month after treatment.
Discomfort or pain in the chest after meals	62/67 (92.5%)	Forty patients (64.5%) reported reduction or complete resolution of symptoms 1 month after treatment.
Flatulence	67/67 (100%)	Forty-seven patients (70%) experienced reduction in flatulence 1 month after treatment.
Frequency of hiccups	Increased frequency: 44/67 (66%)	Twenty-six patients (59%) reported decreased frequency of hiccups 1 month after treatment, while nine (20%) reported increased frequency.
Painful hiccups	59/67 (88%)	Thirty-eight patients (64%) reported a reduction in the number of painful hiccups one month after treatment.
Emetophobia	33/67 (49%)	Two patients (6%) reported symptom improvement at a 1-month follow-up.
Effortful vomiting	43/63 (68%)	Four patients reported not remembering ever having vomited; five patients who experienced effortful vomiting were able to vomit after treatment; and in all cases, the symptom was reported as improved.
Difficulty swallowing solid foods	14/67 (21%)	Twelve patients had sporadic difficulty, one reported frequent difficulty, and one preferentially consumed soft foods. At the 4-month follow-up, two patients (14%) experienced complete resolution.
Difficulty swallowing liquids	14/67 (21%)	All cases were minimal or mild, often occurring during episodes of inattentive drinking; among these 14 patients, 10 (71%) reported sporadic episodes and 4 (29%) reported relatively frequent symptoms. At the 4-month follow-up, 3 patients (21%) showed improvement or complete resolution.
Perception of delayed esophageal transit	34/67 (51%)	Three patients (9%) reported symptom improvement at the 1-month follow-up.
Anxiety symptoms	26/67 (39%)	Thirteen patients (50%) reported improvement and 9 (35%) reported worsening of symptoms at the 1-month follow-up.
Retrosternal heartburn or acid reflux episodes	44/67 (66%)	Nine patients (20%) reported improvement and 11 (42%) reported worsening of symptoms at the 1-month follow-up.
Improvement of symptoms in lying-down position	Improvement: 30/67 (45%); no difference: 21/67 (31%); worsening: 16/67 (24%)	
Symptom progression throughout the day (from morning to evening)	Worsening: 60/67 (90%); no difference: 5/67 (7%); improvement: 2/67 (3%)	
Food and/or drink avoidance	Always: 26/67 (39%); often: 23/67 (34%); sometimes: 8/67 (12%); never/rarely: 10/67 (15%)	Twenty-nine patients (51%) reported a reduction in food avoidance, while 14 patients (25%) no longer avoided any foods at the 4-month follow-up (1-month follow-up not considered due to post-treatment side-effect bias).
Social avoidance	Always: 6/67 (9%); often: 12/67 (18%); sometimes: 16/67 (24%); rarely: 17/67 (25%); never: 16/67 (24%)	Twenty patients (39%) reported reduction in social event avoidance at the 4-month follow-up (1-month follow-up not considered due to post-treatment side-effect bias).
Self-reported global impact on QoL (scale 0–10)	9–10: 35/67 (52.2%); 7–8: 29/67 (43.3%); 5–6: 3/67 (4.4%)	At the 1-month follow-up: 0–4: 29/67 (43.3%); 5–6: 7/67 (10.4%); 7–8: 17/67 (25.4%); 9–10: 14/67 (20.9%).
Smoking	Non-smokers: 53/67 (79%)	
Alcohol	Abstainers: 24/67 (36%); occasional/social drinkers: 42/67 (63%); daily consumption: 1/67 (1.5%)	
At least one first-degree relative affected	8/67 (12%)	

**Table 3 toxins-17-00458-t003:** Clinical data on previous medical evaluations and treatments of the 67 patients enrolled in this study, including the types of specialists consulted, diagnoses received, medications tried, investigations performed, and main pathological findings.

**Medical consultation prior to our care**	47/67 (70%)
**Gastroenterologist**	42/47 (87%)
**General practitioner**	14/47 (30%)
**ENT specialist**	12/47 (25%)
**Others (nutritionist, speech therapist, neurologist, psychiatric, endocrinologist)**	17/47 (36%)
**Given a previous diagnosis**	43/67 (64%)
**Gastroesophageal reflux disease**	22/43 (51%)
**Mood-related disorder/functional symptoms**	14/43 (33%)
**Irritable bowel syndrome**	12/43 (28%)
**Hiatal hernia**	6/43 (14%)
**Gastritis**	4/43 (9%)
**Others (R-CPD *, lactose intolerance, food intolerance, celiac disease, dolicocolon, esophageal spasms, sialhorrea, thyroiditis, constipation)**	40/43 (93%)
**Tried medication before**	42/67 (63%)
**Proton-pump inhibitors**	22/42 (52%)
**Prokinetics**	16/42 (38%)
**Supplements**	11/42 (26%)
**Anti-spasmodic**	6/42 (14%)
**Anti-bloating agents**	6/42 (14%)
**Antiacids/alginates**	6/42/14%)
**Laxatives**	4/42 (10%)
**Anti-depressants/anxiolytics**	4/42 (10%)
**Antibiotics**	4/42 (10%)
**Others**	7/42 (16%)
**Had improvement with medication**	5/67 ** (7.5%)
**Had at least one diagnostic examination**	48/67 (72%)
**EGDS**	37/48 (72%)
**Esopagheal manometry**	18/48 (27%)
**FEES**	12/48 (18%)
**Barium swallow**	10/48 (15%)
**Esophagus pH impedenzometry**	7/48 (10%)
**Abdominal ultrasound**	7/48 (10%)
**Others (e.g., colonoscopy, EMG, videofluoroscopy, CT/MRI of abdomen/brain)**	7/48 (10%)
**Pathological findings at instrumental assessment**	
**EGDS (n = 37)** **Esophagitis (8)** **Gastritis (8)** **Hiatal hernia (3)** **Incompetent cardia (2)** **Biliary reflux (1)**	8/37 (22%) 8/37 (22%) 3/37 (8%) 2/37 (5%) 1/37 (3%)
**Esophageal manometry (n = 18)** **Ineffective esophageal motility (8)** **Hypertonic UES (3)** **Positive provocative test consistent with R-CPD (4)**	8/18 (44%) 3/18 (17%) 4/18 (22%)
**FEES (n = 12)** **UES hypodistension (1)**	1/12 (8%) 1/12 (8%)
**Barium swallow (n = 10)** **Slow esophageal transit (1)** **Pharyngocele (1)**	1/10 (10%) 1/10 (10%)

Abbreviations: **EGDS**: esophagogastroduodenoscopy; **FEES**: fiber-optic endoscopic evaluation of swallowing; **UES**: upper esophageal sphincter; **EMG**: electromyography; **CT/MRI**: computed tomography/magnetic resonance imaging. * Only one of the patients had received a formal diagnosis of R-CPD by a previously consulted physician. ** All experienced only some degree of symptom attenuation without resolution.

**Table 4 toxins-17-00458-t004:** This table summarizes the results of the Friedman test and post hoc analyses assessing changes in individual symptom severity and quality of life (QoL) measures across three timepoints: baseline, 1-month follow-up, and 4-month follow-up (*n* = 40).

	Mean ± SD (Baseline)	Mean ± SD (1 m)	Mean ± SD (4 m)	Friedman χ^2^	*p*-Value	Avg Rank (Baseline)	Avg Rank (1 m)	Avg Rank (4 m)
Ability to burp (0–3)	2.82 ± 0.38	1.70 * ± 1.09	1.70 * ± 1.32	31.6	0.00001	2.54	1.67 *	1.79 *
Burp frequency (0–3)	2.72 * ± 0.45	1.22 * ± 1.17	1.60 * ± 1.35	37.7	0.00001	2.60	1.49 *	1.91 *
Gurgling noises (0–4)	3.5 ± 0.71	2.1 * ± 0.30	2.0 * ± 1.38	41.6	0.00001	2.69	1.65 *	1.66 *
Bloating (0–4)	2.87 ± 1.16	1.45 * ± 1.32	1.55 * ± 1.36	39.8	0.00001	2.64	1.62 *	1.73 *
Chest pain or discomfort (0–4)	2.45 ± 1.04	1.35 * ± 1.29	1.40 * ± 1.26	25.6	0.00001	2.50	1.73 *	1.76 *
Flatulence (0–4)	3.47 ± 0.78	2.07 * ± 1.42	2.30 * ± 1.50	33.3	0.00001	2.57	1.56 *	1.86 *
Hiccup frequency (0–3)	0.9 ± 0.84	0.47 * ± 0.75	0.45 * ± 0.67	10.34	0.006	2.26	1.86 *	1.87 *
Painful hiccups (0–3)	1.7 ± 0.91	0.72 * ± 0.85	0.6 * ± 0.81	37.2	0.00001	2.61	1.71 *	1.67 *
8-item total score (0–28)	20.5 ± 2.86	11.0 * ± 6.9	11.6 * ± 8.0	44.8	0.00001	2.79	1.44 *	1.77 *
Food avoidance (0–4)	3.02 ± 1.05	1.70 * ± 1.42	1.62 * ± 1.50	25.3	0.00001	2.52	1.77 *	1.70 *
Social avoidance (0–4)	1.62 ± 1.30	0.75 * ± 1.06	0.72 * ± 1.11	28.2	0.00001	2.47	1.79 *	1.74 *
Impact on QoL (0–10)	8.42 ± 1.17	4.87 * ± 3.47	5.10 * ± 3.39	34.2	0.00001	2.63	1.67 *	1.69 *
3-item QoL total score (0–18)	13.18 ± 2.50	7.70 * ± 5.00	7.45 * ± 5.41	41.8	0.00001	2.67	1.77 *	1.67 *

Avg Rank: Average rank for each timepoint (baseline, 1 month, 4 months) used in the Friedman test. Asterisks (*) denote statistically significant differences from baseline (*p* < 0.0167, Bonferroni-corrected Wilcoxon signed-rank test).

**Table 5 toxins-17-00458-t005:** Median changes (delta improvement values) in symptom and quality-of-life (QoL) scores from baseline to 4-month follow-up by dose group, stratified by sex.

	Group	Median Δ (10–20 U) [IQR]	Median Δ (25–30 U) [IQR]	*p*-Value
Ability to burp	All patients	0 [0; 2.5]	1.5 [0; 2.5]	0.21
	Females only	0 [0; 2]	2 [1; 2]	0.13
	Males only	0 [0; 3]	0 [0; 3]	0.92
Burp frequency	All patients	0 [0; 2.5]	1.5 [0; 3]	0.22
	Females only	0 [0; 2]	2 [1; 3]	0.10
	Males only	0.5 [0; 3]	0 [0; 2]	0.97
Gurgling noises	All patients	1 [0; 2]	1.5 [0; 3]	0.09
	Females only	0 [0; 2]	3.0 [1; 4]	0.06
	Males only	1 [0; 2]	1 [1; 3]	0.93
Abdominal bloating	All patients	0.5 [0; 2.5]	2 [0; 3]	0.08
	Females only	0 [−1; 1]	2 [1; 3]	0.02 *
	Males only	1.5 [0; 3]	2 [1; 3]	0.93
Chest pain or discomfort	All patients	0.5 [0; 1.5]	1 [0; 2]	0.47
	Females only	1 [0; 2]	1 [0; 2]	0.61
	Males only	0 [0; 1]	1 [0; 2]	0.73
Flatulence	All patients	0 [0; 2]	1 [0; 3]	0.18
	Females only	0 [0; 1]	2 [1; 3]	0.06
	Males only	0.5 [0; 3]	1 [0; 1]	0.90
Hiccup frequency	All patients	0 [0; 1]	0.5 [0; 1]	0.68
	Females only	0 [−1; 1]	1 [0; 1]	0.14
	Males only	0 [0; 1]	0 [−2; 1]	0.27
Painful hiccups	All patients	0 [0; 2]	1 [0; 2]	0.08
	Females only	0 [0; 2]	1 [1; 2]	0.10
	Males only	0.5 [0; 2]	1 [1; 2]	0.47
8-item total score	All patients	1.5 [0; 16]	10.5 [0; 17.5]	0.09
	Females only	1 [0; 10]	13 [9; 19]	0.04 *
	Males only	2 [0; 19]	3 [2; 17]	0.88
Food/beverage avoidance	All patients	1 [0; 3]	1.5 [0; 3]	0.63
	Females only	0 [0; 2]	3 [1; 3]	0.26
	Males only	1 [0; 3]	1 [0; 2]	0.52
Social avoidance	All patients	0 [0; 1.5]	1 [0; 1.5]	0.67
	Females only	0 [0; 1]	1 [0; 2]	0.32
	Males only	0 [0; 2]	0 [0; 1]	0.60
Impact on QoL	All patients	2.5 [0; 6]	3 [0; 7]	0.57
	Females only	0 [0; 6]	5 [3; 7]	0.13
	Males only	3 [0; 6]	1 [0; 2]	0.25
3-item QoL total score	All patients	3.5 [0; 10]	5 [0; 11]	0.59
	Females only	1.5 [0; 8]	7.5 [5; 13]	0.21
	Males only	5 [0; 11]	2 [0; 3]	0.37
Treatment satisfaction	All patients	4.5 [0; 7.5]	7.5 [0; 10]	0.14
	Females only	3 [0; 7]	8 [6; 10]	0.03 *
	Males only	5.5 [0; 8]	4 [1; 7]	0.62

Delta values were calculated as baseline scores minus 4-month scores, so positive values indicate symptom improvement. Data are presented as medians [interquartile ranges]. *p*-values refer to Mann–Whitney U tests comparing the 10–20 U and 25–30 U groups within each subgroup (all patients, females only, males only). * *p* < 0.05 (statistically significant).

**Table 6 toxins-17-00458-t006:** Reported adverse effects and their frequencies among the 67 patients who completed the 1-month follow-up.

Adverse Effect	% of Subjects
Dysphagia	56/67 (83%)
Hoarseness	17/67 (25%)
Acid reflux	12/67 (18%)
Exertional dyspnea	4/67 (6%)
Injection site pain > 24 h post-treatment	2/67 (3%)

**Table 7 toxins-17-00458-t007:** Summary of electromyographic parameters of the cricopharyngeus muscle recorded at baseline in the full cohort.

	Electrophysiological Parameter	Median (IQR)
Tonic basal activity	Mean amplitude (µV)	482 (322–671)
	Peak amplitude (µV)	718 (440–858)
	Area under the curve (µV·sec)	516 (315–685)
Foreburst	Mean amplitude (µV)	663 (410–819)
	Peak amplitude (µV)	802 (519–1024)
	Area under the curve (µV·sec)	635 (406–793)
	Duration (s)	0.82 (0.55–1.35)
Cricopharyngeus EMG pause	Mean amplitude (µV)	178 (132–213)
	Peak amplitude (µV)	226 (181–290)
	Area under the curve (µV·sec)	161 (107–203)
	Duration (s)	0.57 (0.40–0.70)
Squeezing	Mean amplitude (µV)	673 (549–903)
	Peak amplitude (µV)	976 (737–1271)
	Area under the curve (µV·sec)	756 (589–949)
	Duration (s)	1.32 (1.00–1.71)

Values are expressed as medians and interquartile ranges (IQRs).

## Data Availability

The raw data used in this study will be available in the Zenodo repository at 10.5281/zenodo.16082712.

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
