# Peer review of "Unilateral EMG-Guided Botulinum Toxin for Retrograde Cricopharyngeus Dysfunction: A Prospective Clinical and Neurophysiological Study"

_toxins, 2025, doi:10.3390/toxins17090458_

Round 1
Reviewer 1 Report
Comments and Suggestions for Authors
The manuscript reports the results of an innovative exploratory study into the treatment of retrograde cricopharyngeus dysfunction, a recently described nosological entity characterised by excessive contraction of the cricopharyngeus muscles and an inability to burp, using botulinum toxin. The manuscript is clearly written and reports consistent data. Using a neurophysiological approach during injection offers significant advantages in terms of the procedure's safety and accuracy. I suggest a few minor improvements:
- Table 1: The questionnaire should be provided as a separate PDF in the supplementary material.
- Section 2.2: Why was the reported benefit of botulinum toxin injection higher during the first week after treatment (92.5% of patients) and lower after one month (64.4–26.4% responder rates)? A higher benefit should be expected after one month, given that the effect of botulinum toxin is considered maximal at two to four weeks after injection. Please elaborate on this point.
- Table 5: Please specify in the table header that the delta change reported in the table is actually a 'delta improvement', as clarified later.
- Botulinum toxin indication: R-CPD is not among the approved indications for botulinum toxin treatment. The authors should clarify how the treatment was formally justified (off-label based on published literature? Was R-CPD considered a type of cricopharyngeal muscle dystonia?
Author Response
The manuscript reports the results of an innovative exploratory study into the treatment of retrograde cricopharyngeus dysfunction, a recently described nosological entity characterised by excessive contraction of the cricopharyngeus muscles and an inability to burp, using botulinum toxin. The manuscript is clearly written and reports consistent data. Using a neurophysiological approach during injection offers significant advantages in terms of the procedure's safety and accuracy. I suggest a few minor improvements:
- Table 1: The questionnaire should be provided as a separate PDF in the supplementary material.
Answer: We sincerely thank the reviewer for carefully reading and evaluating our manuscript, and for providing constructive and insightful comments that have helped us to improve the quality and clarity of the work.
Regarding the first point, the complete questionnaire has now been provided as a separate PDF file (Supplementary Material 1). In the main text, Table 1 has been replaced by a more concise summary version, reporting only the domains and items explored.
- Section 2.2: Why was the reported benefit of botulinum toxin injection higher during the first week after treatment (92.5% of patients) and lower after one month (64.4–26.4% responder rates)? A higher benefit should be expected after one month, given that the effect of botulinum toxin is considered maximal at two to four weeks after injection. Please elaborate on this point.
Answer: We thank the reviewer for raising this important point. The apparent discrepancy arises from the fact that a structured clinical evaluation with the full questionnaire was performed only at the 1-month follow-up, not during the first post-treatment week. The figure of 92.5% refers exclusively to the restoration or facilitation of the ability to burp, which was reported by most patients within a few days after injection, even though the effect was often incomplete or not yet perceived as satisfactory. We considered this parameter relevant, as it provides an objective early marker of treatment effect on cricopharyngeus relaxation and indirectly supports the diagnosis of R-CPD, in line with previous reports in the literature. In contrast, the 1-month responder rate was based on the more stringent criterion of overall clinical benefit (satisfaction score ≥6/10), which reflects not only the ability to burp but also the persistence and impact of symptom improvement on daily life. Therefore, the early high percentage indicates the initial pharmacological effect on the muscle, while the lower responder rate at one month reflects that in some patients this effect was not sufficient to translate into a clinically meaningful or durable improvement across the broader symptom spectrum.
The corresponding text in the Discussion has been modified as follows:
Comparisons across studies remain challenging due to variations in follow-up duration, outcome definitions, injection techniques, and toxin formulations. In our cohort, 92.5% of patients regained the ability to burp within the first week after treatment, in line with the findings of Lichien et al. [7]. This parameter mainly reflected the early pharmacological effect of BoNT on cricopharyngeus relaxation and was often incomplete or not yet perceived as satisfactory. Nevertheless, this early improvement is clinically relevant, as it supports the diagnosis of R-CPD. However, it did not always translate into a broader and sustained benefit across the full symptom spectrum, which explains the lower responder rate observed at the 1-month evaluation when assessed through the structured questionnaire.
- Table 5: Please specify in the table header that the delta change reported in the table is actually a 'delta improvement', as clarified later.
Answer: As suggested, we have clarified in the table header that the reported delta values represent delta improvement. In addition, the explanatory note specifying that positive values indicate symptom improvement has been moved from the bottom of the table into the legend, so that all clarifications are presented in a single, easily accessible section.
- Botulinum toxin indication: R-CPD is not among the approved indications for botulinum toxin treatment. The authors should clarify how the treatment was formally justified (off-label based on published literature? Was R-CPD considered a type of cricopharyngeal muscle dystonia?
Answer: We thank the reviewer for this relevant comment. We confirm that botulinum toxin injections for R-CPD were performed as off-label use, justified by the growing body of literature supporting their efficacy and safety. All patients provided specific written informed consent for the off-label procedure. At present, the pathophysiology of R-CPD is not fully understood, and we do not hypothesize mechanisms similar to dystonia. Rather, R-CPD may represent a local motility disorder of the cricopharyngeus muscle, in some cases possibly the expression of a broader esophageal motility disturbance, as suggested by the frequent association with ineffective esophageal motility (IEM). The contribution of central (brainstem or supratentorial) mechanisms remains to be clarified.
The following information has been added within the text (treatment procedures paragraph): “The procedure was performed as an off-label use, justified by available literature data and carried out after obtaining specific informed consent for off-label treatment”.
Reviewer 2 Report
Comments and Suggestions for Authors
This work aimed to determine whether two types of doses of botulinum neurotoxin had different effects on retrograde cricopharyngeus dysfunction. The results provide some useful information. I would like to offer several suggestions:
- It is not suitable to use correlation analysis to “predict” a result. A regression analysis should be used to quantify the causal relationship and clarify the magnitude and direction (positive/negative) of the influence of the independent variable on the dependent v After establishing a regression equation, inputting independent variable data can calculate the predicted value - this is a prediction; when using multiple regressions, confounding factors can be controlled and thus the relationship between the target variable and the result can be more accurately reflected.
- For cricopharyngeus dysfunction caused by neurological diseases, EMG signals on the right side may be different from those on the left side. For EMG data in this study, there are any differences in EMG activities between the two sides?
- During injection, were the syringe needles covered by electrical insulators? Otherwise, the needle body and tip may touch/be inside in two different muscles, which will contaminate EMG signals.
Author Response
Responses to Reviewer 2
This work aimed to determine whether two types of doses of botulinum neurotoxin had different effects on retrograde cricopharyngeus dysfunction. The results provide some useful information. I would like to offer several suggestions:
It is not suitable to use correlation analysis to “predict” a result. A regression analysis should be used to quantify the causal relationship and clarify the magnitude and direction (positive/negative) of the influence of the independent variable on the dependent v After establishing a regression equation, inputting independent variable data can calculate the predicted value - this is a prediction; when using multiple regressions, confounding factors can be controlled and thus the relationship between the target variable and the result can be more accurately reflected.
Answer: We sincerely thank the reviewer for the time dedicated to carefully evaluating our manuscript and for the constructive comments that helped us improve the robustness of the analyses and the clarity of the results. We fully agree with the observation that correlation analysis alone is not suitable to predict outcomes. Following the reviewer’s suggestion, we performed logistic regression analyses considering treatment response (yes/no) as the dependent variable and sex, age, and dose as independent covariates. In univariate analyses, sex and age were not significantly associated with treatment outcome, and dose showed only a non-significant trend. However, when including electrophysiological parameters in multivariate models (adjusted for sex, age, and dose), several neurophysiological measures emerged as significantly associated with treatment response, namely cricopharyngeus EMG pause peak amplitude, pause mean amplitude, pause area under the curve, and squeezing mean amplitude. Noteworthy, dose itself reached statistical significance only in models that also included either cricopharyngeus EMG pause peak amplitude or squeezing mean amplitude, despite not being significant in the univariate setting. These results indicate that both BoNT dose and specific electrophysiological features independently contribute to predicting treatment response. The revised analyses and interpretations have been incorporated into the main text, with changes highlighted in the Statistical Analysis, Results, Discussion, and Conclusions sections.
For cricopharyngeus dysfunction caused by neurological diseases, EMG signals on the right side may be different from those on the left side. For EMG data in this study, there are any differences in EMG activities between the two sides?
Answer: We thank the reviewer for this insightful comment. As noted, almost all injections and EMG recordings were performed on the right cricopharyngeus muscle, with the exception of a small subgroup of patients (n = 6) in whom the left side was used due to technical difficulties in clearly identifying the right portion of the muscle. Although the limited number of left-sided cases precludes any robust statistical comparison, we performed an exploratory t-test between right- and left-sided recordings. This analysis did not reveal any significant differences or trends across all EMG parameters. We have now included this information in the revised manuscript.
During injection, were the syringe needles covered by electrical insulators? Otherwise, the needle body and tip may touch/be inside in two different muscles, which will contaminate EMG signals.
Answer: We thank the reviewer for this important point. We used Ambu Neuroline Inoject needle electrodes (0.50 mm, 25 G). These needles consist of a stainless‑steel shaft coated with a silicone layer and featuring an underlying acrylic insulating layer, such that only the beveled tip is electrically active, while the shaft remains fully insulated. This design effectively prevents electrical contamination from adjacent muscles. We have clarified this detail in the Methods section of the revised manuscript as follows: “We used Ambu Neuroline Inoject needle electrodes (0.50 mm, 25 G). These needles have a stainless-steel shaft covered by an acrylic insulating layer and a silicone outer coating, so that only the beveled tip is electrically active while the shaft remains fully insulated, thereby minimizing the risk of EMG signal contamination”.